# Enhanced Oil Recovery with Size-Dependent Interactions of Nanoparticles Surface-Modified by Zwitterionic Surfactants

Han Am Son [1] and Taehun Lee [2],*

1   Department of Energy Resources Engineering, Pukyong National University, Busan 48547, Korea;
    hason@pknu.ac.kr
2   Oil and Gas Research Center, Petroleum and Marine Research Division, Korea Institute of Geoscience and
    Mineral Resources, Daejeon 34132, Korea
*   Correspondence: thlee@kigam.re.kr; Tel.: +82-42-868-3076

**Abstract:** This study reports the size-dependent interactions of silica nanoparticle (NP) dispersions with oil, which facilitate oil recovery from sandstone rock. Herein, we studied various 7–22 nm sized colloidal silica NPs (CSNPs; the colloidal state when dispersed in aqueous solutions) and fumed silica nanoparticles (FSNPs; the dry powder state). Interfacial tension at the oil-nanofluids interface declined with decreasing NP size in a range from 7 to 22 nm. This is because NP spatial density at the interface increased with smaller particle size, thereby, the interface area per NP decreased to approximately 1/30, and interfacial energy had reduced enough. In addition, smaller NPs more strongly were adsorbed to the rock because of improved diffusion in suspension and increased adsorption density. This caused creating a wedge film between oil and rock, which changed the oil contact angle. Due to this effect, core flooding experiments indicated that oil recovery increased with decreasing particle size. However, FSNP dispersions exhibited low recovery factor because of particle aggregation. This phenomenon may facilitate massive permeability reduction, thus causing oil trapping inside rock pore. We found that both the sizes and types of CSNPs and FSNP affected the Interfacial tension at oil-water interface and rock surface wettability, which influenced ultimate oil recovery.

**Keywords:** enhanced oil recovery; nanoparticle size; colloidal silica nanoparticle (CSNP); fumed silica nanoparticle (FSNP)

## 1. Introduction

Recently, nanotechnological applications have attracted great attention in the context of oil reservoirs with small pores and pore throats [1–8]. Nanoparticle (NP) dispersions change the wettability of porous media. Wettability controls the fluid flow, thus affecting oil recovery during water flooding. Many studies have shown that NPs render rock surfaces more hydrophilically wettable [9–12]. In addition, NPs reduce interfacial tension (IFT) when they adsorb to oil–water interfaces [13–15]. This greatly improves displacement efficiency, and thus, oil recovery. However, in many studies regarding enhanced oil recovery using NPs, NP stability remained problematic [16,17]. The low stability of individual particles in a high-saline environment triggers aggregation, which may obstruct pores and possibly damage the rock formation [17,18]. Initially, small clusters form, followed by larger clusters, reflecting the balance of forces according to the Derjaguin–Landau–Verwey–Overbeek theory. The gradual increase in cluster size causes undesirable NP sedimentation in a porous medium, separating the NPs from the nanofluid [19]. High temperature enhances Brownian motion and the consequent probability of particle collision; oppositely charged ions in solution limit the electrostatic repulsion between particles, and thus, promote flocculation [17]. Therefore, particle surface modification is essential when preparing thermally and kinetically stable NP dispersions. To this end, NPs are blended with surfactants (stabilizers) [20,21]. The synergistic effects of surfactants and NPs have been widely

researched; these lower the IFT and change the surface wettability of reservoir rock [22,23]. Surfactant–NP systems have attracted attention because of their increased particle stability, reduced NP dosage, decreased surfactant adsorption loss, and enhanced efficiency [17,24]. As well as the surfactant–NP system, NP surface modifications using a polymer chain improve particle stability. Yang et al. [25] found that PAA (polyacrylic acid)-modified nanoparticles were more dispersible in aqueous solutions compared with unmodified particles. They concluded that PAA-modified particles flowed approximately 10-fold longer through a silica sand column compared with unmodified particles. Bagaria et al. [26] also modified NP surfaces by copolymerization with PSS (polystyrene sulfonate); the modified particles formed stable colloids in high-concentration brine. In our previous study, silica NP stability was enhanced after absorption of a zwitterionic surfactant and PSS-*co*-MA [27]. We found that the adsorption of PSS-*co*-MA on silica NPs was possibly generated by van der Waals attraction. Furthermore, adsorption of zwitterionic surfactant onto NPs is mediated by dipole-charge interaction between the trimethylammoniun groups of zwitterionic surfactant and negatively charged particles with PSS-*co*-MA under brine. Previous study showed that our surface-modified could reduce the IFT and enhance oil recovery. For further research, this study focused on the effect of nanoparticle size on interfacial tension and rock wettability.

Size variation of NPs strongly affects their physical properties. For example, NP size can affect wettability. Wasan and Nikolov [28,29] showed that the pressure caused by NP deposition at oil–water–solid contact regions (i.e., disjoining pressure) is the primary mechanism of oil displacement in sandstone. They concluded that disjoining pressure is related to the size of the NP because the energy at oil—water–solid contact regions is a function of the particle diameter. However, in contrast, Al-Anssari et al. [30] found that the NP size (5 and 25 nm) did not influence the wettability alteration of the oil–wet calcite rocks, indicating that there was no difference in particle adsorption on the rock surface. These previous studies have shown different results. Further experimental studies are needed to determine the effect of the size of NP on rock wettability. NP number density and diffusion of particles in suspension may change depending on the particle size, which can change rock wettability. Thus, this study focused on the diffusion of NPs and NP number density in suspension for surface wettability.

In addition, there is a serious lack of information in terms of how nanoparticle size influences IFT at the oil–water interface. According to previous studies, Saad and Li [31] and Brown et al. [32], showed that large NPs exhibit higher surface tension. They concluded that this trend is likely due to the strengthened van der Waals force between particles as particle size increases. However, Kutuzov [33] found that IFT decreased with increasing particle size. They concluded that the rate of adsorption of the nanoparticles at the oil–water interface decreases with decreasing nanoparticle size. Manga et al. [34] also found that the effects of very small NPs (<20 nm) on IFT are less clear. They concluded that in cases of small nanoparticle adsorption, the resulting behavior is difficult to predict. Thus, previous research showed that the influence of particle adsorption on the oil–water interfacial tension is not well understood.

For research on the variation of IFT and wettability, the NP spatial density at the oil–water interface is important in terms of interfacial stability. Nonetheless, the relationship of IFT reduction as a function of the interfacial area covered by differently sized NP and the desorption energy required to remove one NP has not been clearly elucidated. Thus, unlike the previous study, we tried to analyze how the size of NPs affects the interfacial area at the oil–water interface covering per NP, and, thereby, whether interfacial tension is affected. Here, we evaluated colloidal silica NPs (CSNPs) of various sizes (7, 12, and 22 nm), and a fumed silica NP (FSNP) to which the zwitterionic surfactant PSS-*co*-MA had been adsorbed (attracted by the negative charges of silica). Specifically, we focused on how NP size and type affected the IFT of the oil/nanofluid interface and rock surface wettability. To explore how NP density affected IFT, we calculated the interface areas covered by NPs, assuming that all NPs were located at the interface. We studied the effects of particle size on

wettability by observing changes in the contact angles according to NP size in three-phase rock–oil–water systems. We also investigated the flow within porous rocks according to particle size and type in terms of oil recovery; we performed core flooding experiments using Berea sandstone.

## 2. Materials and Methods

### 2.1. Materials

The three hydrophilic CSNP suspensions used were Ludox SM (30 wt% suspension in $H_2O$, particle diameter 7 nm), Ludox HS-40 (40 wt% suspension in $H_2O$, particle diameter 12 nm), and Ludox CL-X (45 wt% suspension in $H_2O$, particle diameter 22 nm) (all from Sigma-Aldrich, St. Louis, MO, USA). They were delivered in a colloidal state dispersed in an aqueous phase, of purity > 99.8%. Additionally, hydrophilic FSNPs (dry powder; Aerosil 200, specific surface area approximately 200 $m^2 g^{-1}$ and particle diameter 12 nm) were purchased from Evonik Industries, Germany. An anionic polymer, poly (4-styrenesulfonic acid-co-maleic acid) sodium salt (PSS-*co*-MA) of 99% purity was purchased from Sigma-Aldrich, as was a zwitterion surfactant, 3-(*N*,*N*-dimethylmyristylammonio) propanesulfonate (TPS) of 99% purity. The oil used was KF-96-10 (Shinetsu), with a viscosity of approximately 10 cp. Brine (3 wt% NaCl) served as the aqueous phase. The flow experiments employed Berea sandstone with a diameter of 3.8 cm, length of 5–6 cm, porosity of 19–22%, and permeability of 138–154 md.

### 2.2. Preparation of Nanoparticle Dispersions

First, four aqueous phases were prepared by the addition of the three CSNPs (0.5 wt% silica content, diameters 7, 12, and 22 nm) and FSNP (0.5 wt% silica content, diameter 12 nm) to brine. To surface-modify the NPs, PSS-*co*-MA (an anionic polymer) was added to 0.5 wt% of each solution and the mixtures were stirred for 5 min with a magnetic stirrer operating at 800 rpm [27]. Next, TPS (a zwitterion surfactant) was added to 0.1 wt% followed by mixing for approximately 1 h and then for 2 min in 2 s/1 s on-off bursts at 3000 rpm, using an ultrasonic homogenizer (Digital Sonifier, Branson, MA, USA).

### 2.3. Characterization

A transmission electron microscope (TEM; JEN 2100F, Jeol, CO, USA) was used to observe NPs. The NP hydrodynamic sizes and size distributions in solution were analyzed via DLS (ZS-90, Malvern, OH, USA); the ζ potentials were also measured. The suspension concentrations were 10.0 mg/mL; measurements were collected over 3 min at 25 °C. The steady-state viscosity of each colloidal dispersion was measured over a range of shear rates at 25 °C, using a Brookfield DV3 instrument. IFT and contact angle measurements were performed on solutions stabilized with NPs and surfactant. We used a Theta tensiometer (Biolin Scientific, Västra Frölunda, Sweden) for these measurements; data were analyzed with the aid of One Attension software. All measurements were performed at 25 °C.

### 2.4. Evaluation of Emulsification

The addition of 4 mL of oil (viscosity 10 cp) to each NP dispersion produced microscale emulsions after application of mechanical stress at room temperature. Each sample was mixed for 2 min in 2 s/1 s on-off bursts at 3000 rpm, using the ultrasonic homogenizer. Phase separation was evaluated by addition of 8-mL amounts of emulsions to 10-mL bottles, followed by storage for 10 days.

### 2.5. Core Flooding Experiments

The core flooding apparatus featured an injection pump (500D Syringe Pump, Teledyne ISCO, Lincoln, OR, USA), an accumulator (CFR-100-100, TEMCO Inc., Loveland, CO, USA), a core holder (Young Sung Tech., Daejeon, Korea), and a measuring cylinder (Figure 1). The flow line from the accumulator was connected to the core holder and packed with Berea sandstone. Water was injected into the sandstone, followed by oil, until

no more water flowed. The initial water and oil volume fractions were calculated from the amounts of water produced. In the core flooding experiments, various CSNP and FSNP dispersions (silica NPs, the PSS-*co*-MA polymer, and TPS in brine) were injected into the core holder through the accumulator at a constant rate of 1 mL min-1. The fluid that flowed through the sandstone was collected in a measuring cylinder. Oil recovery was calculated by reference to the proportions of extracted oil (of the original oil). Pressure transducers (DXD, Heise, KS, USA) were installed at the inlet and outlet of the core holder. All experiments were performed at 25 °C and the outlet pressure of the core holder for experiments was atmospheric pressure (14.7 psi).

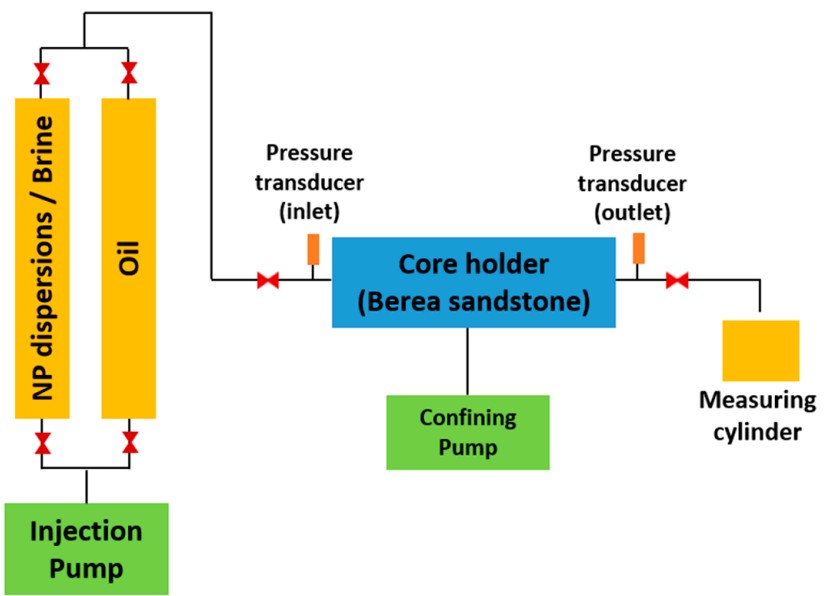

**Figure 1.** The diagram of the core flooding system.

## 3. Results and Discussion

### 3.1. Dispersion of Silica NPs in Aqueous Solutions

We measured the sizes and electric potentials of silica NPs (7 nm CSNP, 12 nm CSNP, 12 nm FSNP, and 22 nm CSNP) in aqueous solutions. TEM images confirmed that the sizes were as expected; aggregation was lacking (Figure 2). When NPs were suspended in brine, the mean DLS diameters were approximately 10, 15, and 25 nm, respectively (Figure 3a). The sizes in dispersions with surfactant and polymer were similar to those of bare NPs (Figure 3b). Thus, aggregation was absent; the dispersions were thermodynamically stable and homogeneous in the presence of surfactant and polymer, attributable to the negatively charged surfaces of silica NPs [35]. The zeta potentials showed that stability was attributable to the constant distance between NPs at the oil–water interface; particle–particle repulsion was in effect (Figure 3c). However, the bare FSNPs were approximately 12 nm in diameter, but formed undispersed (aggregated) clusters attributable to incomplete dispersion or physical agglomeration during drying, although the $\zeta$ potential was less than $-38$ mV (Figure 3c). Thus, regardless of dispersion in water, the FSNPs remained agglomerated, unlike the CSNPs.

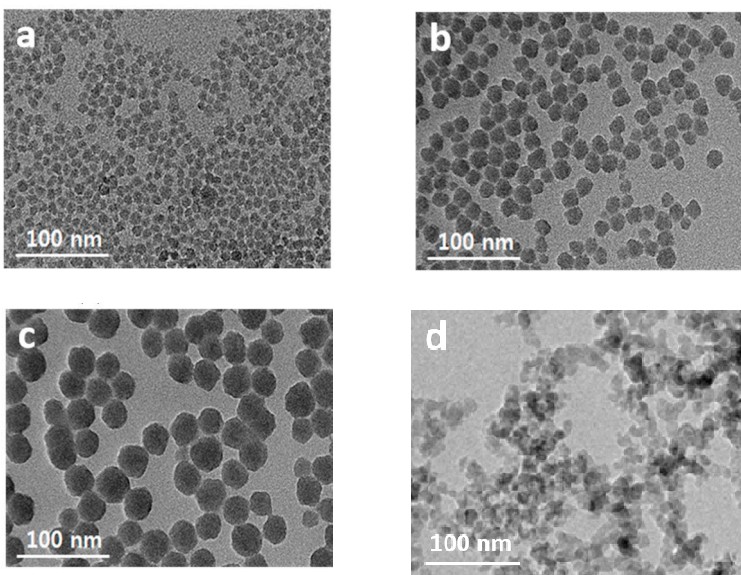

**Figure 2.** TEM images of differently sized silica particles. (**a**) 7 nm CSNPs; (**b**) 12 nm CSNPs; (**c**) 22 nm CSNPs; (**d**) 12 nm FSNPs

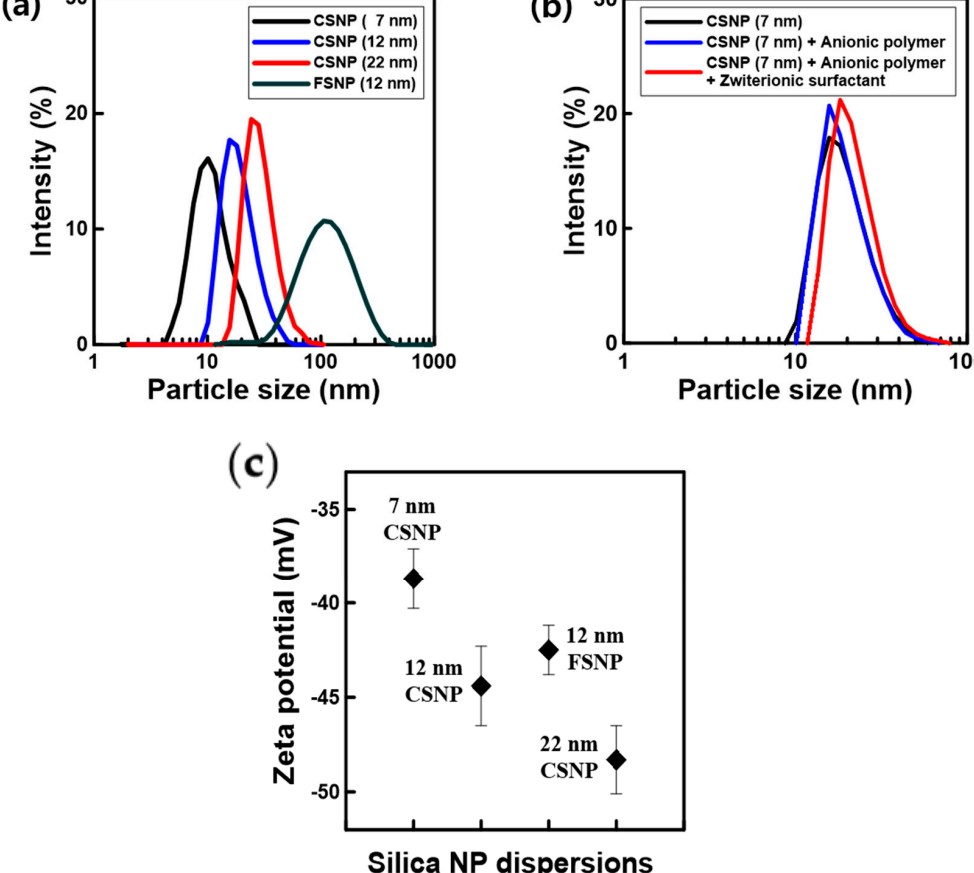

**Figure 3.** (**a**) DLS analysis of differently sized silica NPs, (**b**) DLS analysis of 7 nm with anionic polymer and zwitterionic surfactant, and (**c**) zeta potential analysis of differently sized silica NPs in colloidal dispersions.

### 3.2. Interfacial Stability of NPs at the Oil–Water Interface

To enhance interfacial stability, NPs must adhere strongly to the oil–water interface. Under such conditions, NP layers form around oil drops, preventing their close approach, and thus, prohibiting coalescence [36]. As particle size increases, the oil–water interface behavior improves because adhesion energy is enhanced. In principle, the adhesion energy is influenced by the IFT, as well as NP wettability and size. This relationship is demonstrated as follows: $E = \pi R^2 \gamma_{ow} (1 - \cos \theta)^2$, where $E$ is the adhesion energy of a particle, $R$ is the particle radius, $\gamma_{ow}$ is the IFT, and $\theta$ is the contact angle of the particle at the oil–water interface. Thus, small particles exhibit lower adhesion energy, compared with large particles. We found that all silica NPs remained in stable emulsions for ten days, and thus, readily adsorbed to the oil–water interface, this included 7 nm CSNPs (Figure 4).

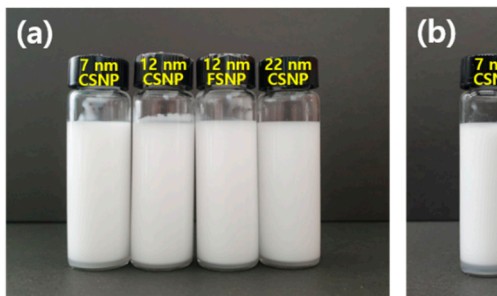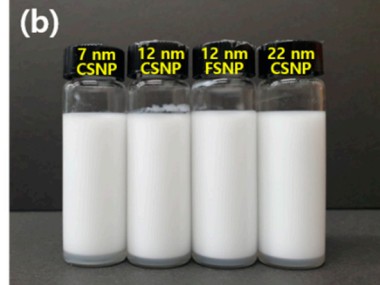

**Figure 4.** Phase stability tests of emulsions stabilized by differently sized NPs (**a**) immediately upon emulsion formation and (**b**) after ten days.

To evaluate oil droplet sizes in emulsions, we performed bright-field microscopy (Figure 5a). Regardless of particle size, all droplets ranged in size from 0.5 to 4 μm and the mean diameter of all droplets ($D_{drop.}$) ranged from 1.47 μm (12 nm CSNP) to 1.76 μm (12 nm FSNP) (Figure 5b). As shown in Table 1, the mean $D_{drop.}$ was used to calculate the mean droplet volume ($V_{drop.} = 1/6\ \pi \times D_{drop.}^3$). All emulsions included $4.0 \times 10^{12}$ μm$^3$ of oil ($V_{tot.\ oil\ vol.}$). Thus, we calculated the total numbers of oil droplets ($N_{total\ drop.} = V_{tot.\ oil\ vol.}/V_{drop.}$) and the surface area per droplet ($S_{aver.\ drop.} = \pi \times D_{drop.}^2$). The $V_{drop.}$ and $S_{aver.\ drop.}$ decreased as drop size ($D_{drop.}$) decreased. Because $V_{tot.\ oil\ vol.}$ was identical in each instance, $N_{total\ drop.}$ increased as droplet size decreased. We found that the 12 nm CSNPs had the smallest droplet diameter, and thus, formed the largest number of droplets (Table 1).

**Table 1.** Mean surface area per droplet.

| | 7 nm CSNP | 12 nm CSNP | 22 nm CSNP | 12 nm FSNP |
|---|---|---|---|---|
| Mean droplet diameter ($D_{drop.}$), μm | 1.63 | 1.47 | 1.66 | 1.76 |
| Mean surface area of droplet ($S_{aver.\ drop.} = \pi \times D_{drop.}^2$, μm$^2$) | 8.34 | 6.78 | 8.66 | 9.73 |
| Mean droplet volume ($V_{drop.} = 1/6\ \pi \times D_{drop.}^3$), μm$^3$ | 2.26 | 1.66 | 2.39 | 2.85 |
| Total oil volume in emulsion ($V_{tot.\ oil\ vol.}$), μm$^3$ | $4.0 \times 10^{12}$ | | | |
| Total number of emulsion droplets ($N_{tot.\ drop.} = V_{tot.\ oil\ vol.}/V_{drop.}$) | $1.76 \times 10^{12}$ | $2.40 \times 10^{12}$ | $1.67 \times 10^{12}$ | $1.40 \times 10^{12}$ |
| Total surface area of emulsion droplets ($S_{tot.\ drop.} = N_{tot.\ drop.}\ S_{aver.\ drop.}$), μm$^3$ | $1.47 \times 10^{13}$ | $1.63 \times 10^{13}$ | $1.44 \times 10^{13}$ | $1.36 \times 10^{13}$ |

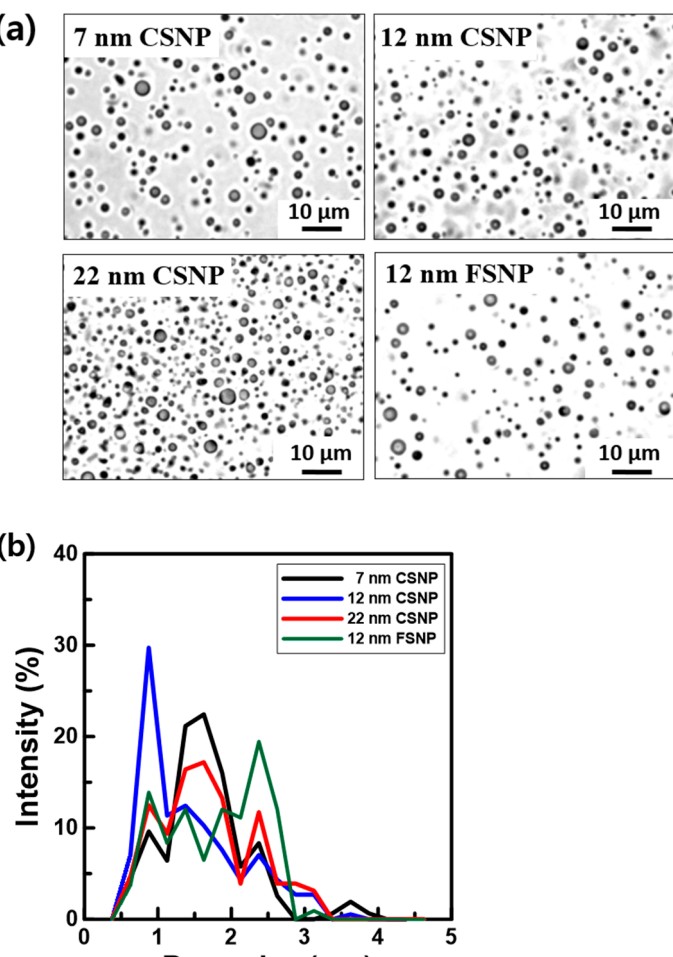

**Figure 5.** (**a**) Droplets of different sizes in NP-stabilized emulsions and (**b**) their size distributions.

### 3.3. Effect of NP Number Density on the Oil–Water IFT

The NP spatial density at the oil–water interface is important in terms of interfacial stability. An increased NP number density reduces the interfacial area covered by each NP. To explore the effect of NP number density on IFT, we calculated the interface areas covered by one NP, assuming that all NPs were located at the oil–water interface. First, we calculated the total number of NPs present in the emulsion ($N_{tot. NP}$) considering the total mass of NPs ($M_{tot. NP}$) and the density of each bare NP ($\rho_{NP}$) used for emulsion formation. As listed in Table 2, $M_{tot. NP}$ for each silica NP was 0.0199 g, thus, 0.5 wt% in the aqueous phase. $\rho_{NP}$ ranged from approximately $0.5 \times 10^{-21}$ (12 nm FSNP) to $2.5 \times 10^{-21}$ g/nm$^3$ (7 nm CSNP). Thus, the total volume of NPs in emulsion ($V_{tot. NP} = M_{tot. NP}/\rho_{NP}$) ranged from approximately $7.93 \times 10^{18}$ nm$^3$ (7 nm CSNP) to $3.98 \times 10^{19}$ nm$^3$ (12 nm FSNP), and the total number of NPs ($N_{tot. NP} = V_{tot. NP}/V_{NP}$) ranged from approximately $1.46 \times 10^{15}$ (22 nm CSNP) to $4.42 \times 10^{16}$ (7 nm CSNP). The interfacial area covered by each NP ($A_{inter. NP}$) was the total surface area of an emulsion droplet ($S_{tot. drop.}$) divided by $N_{tot. NP}$ $A_{inter. NP}$, thus ranged from 309.99 nm$^2$ (12 nm FSNP) to 9836.12 nm$^2$ (22 nm CSNP). The FSNP dispersion has a low density; many particles attain the interface when NPs of the same mass are injected for emulsion generation. Therefore, the interfacial area per particle is low for the FSNP dispersion.

**Table 2.** Interfacial areas covered by single NPs.

|  | 7 nm CSNP | 12 nm CSNP | 22 nm CSNP | 12 nm FSNP |
|---|---|---|---|---|
| Total mass of bare NPs used for emulsion formation ($M_{tot.\ NP}$), g | 0.0199 (0.5 wt% in the aqueous phase) | | | |
| Density of bare NPs ($\rho_{NP}$), g/nm$^3$ | $2.50 \times 10^{-21}$ | $2.36 \times 10^{-21}$ | $2.42 \times 10^{-21}$ | Approximately $0.50 \times 10^{-22}$ |
| Total volume of NPs in emulsion ($V_{tot.\ NP} = M_{tot.\ NP}/\rho_{NP}$), nm$^3$ | $7.93 \times 10^{18}$ | $8.41 \times 10^{18}$ | $8.19 \times 10^{18}$ | $3.98 \times 10^{19}$ |
| Mean diameter of bare NPs ($D_{NP}$), nm | 7 | 12 | 22 | 12 |
| Volume of bare NPs ($V_{NP} = 1/6\ \pi \times D_{NP}{}^3$), nm$^3$ | 179.59 | 904.77 | 5575.27 | 904.77 |
| Total number of NPs in emulsion ($N_{tot.\ NP} = V_{tot.\ NP}/V_{NP}$) | $4.42 \times 10^{16}$ | $9.30 \times 10^{15}$ | $1.46 \times 10^{15}$ | $4.39 \times 10^{16}$ |
| Number of NPs per unit surface area ($N_{sur.\ NP} = N_{tot.\ NP}/S_{tot.\ drop.}$), 1/$\mu$m$^2$ | 3002 | 569 | 101 | 3225 |
| Interfacial area covered per NP ($A_{inter.\ NP} = S_{tot.\ drop.}/N_{tot.\ NP}$), nm$^2$ | 333.10 | 1754.51 | 9836.82 | 309.99 |

We measured the changes in the interfacial properties of oil–water mixtures according to the variation in interfacial area per NP. Assuming that NP–NP interactions can be neglected, the IFT ($\gamma_c$) is $\gamma_c = \gamma_o - \frac{N_P E_d}{A}$, where $\gamma_o$ is the IFT of the bare interface, $E_d$ is the desorption energy required to remove one NP from the interface, and $N_P$ is the number of NPs in a given interfacial area A [13,37]. $\gamma_c$ is the IFT between the aqueous phase containing silica NPs and the oil phase, and $\gamma_o$ is 14 mN/m, yielding the IFT between oil and the aqueous phase in the absence of silica NPs (the aqueous phase contains PSS-*co*-MA and the zwitterionic TPS surfactant) (Figure 6). Thus, any increase in the IFT-difference ($\gamma_o - \gamma_c$) indicates that NPs reduce the IFT ($\gamma_c$). We found that as the interfacial area per NP decreased, the IFT-difference increased (Figure 6a), because the NP–NP spacing became smaller, thereby the interface area per NP decreased to approximately 1/30. We calculated the desorption energy required to remove one NP ($E_d$), this comprised the IFT-difference ($\gamma_o - \gamma_c$) as a function of $E_d$ (Figure 6b). A low $E_d$ indicates that an NP is easily adsorbed to and detached from the interface, therefore, more effectively reducing the IFT at low $E_d$. Although the $E_d$ was indeed low, we confirmed that all CSNP dispersions and the FSNP dispersion exhibited adequate emulsion stability (i.e., no phase separation) (Figure 4). The 12 nm FSNP dispersion adsorbed readily to the oil–water interface at low values of $E_d$ and exhibited stable interface behavior, thereby increasing the IFT-difference ($\gamma_o - \gamma_c$).

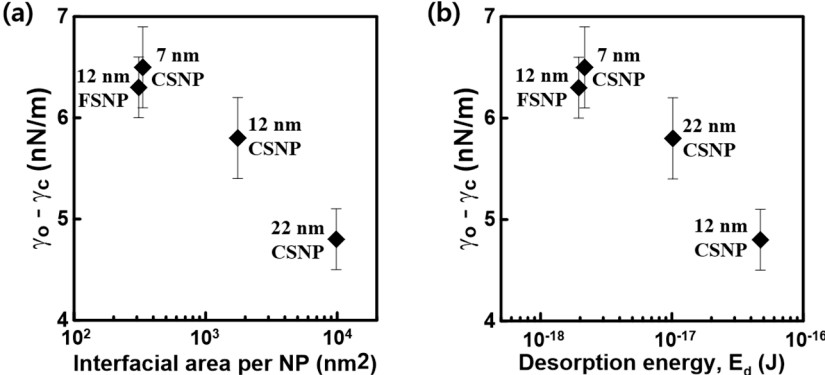

**Figure 6.** (**a**) IFT reduction as function of the interfacial area covered per NP and (**b**) the desorption energy required to remove one NP.

### 3.4. Contact Angles According to NP Size in the Three-Phase Rock–Oil–Aqueous System

We observed changes in contact angle according to NP size in the three-phase rock–oil–aqueous system (Figure 7a). The contact angle was 120.5° when the NPs did not feature an aqueous phase (rock/oil/pure water). However, the contact angle increased to 142° as the CSNP size decreased (Figure 7b), indicating enhanced wettability caused by NP adsorption onto the rock surface. There are two possible reasons for these findings. First, NP diffusion in suspension increases as the particle size falls. The relevant equation is $D = \frac{k_B T}{6\pi\mu a}$; the diffusion coefficient $D$ is inversely proportional to the particle radius $a$. $k_B$ is the Boltzmann constant, $T$ is the absolute temperature, and $\mu$ is the viscosity.

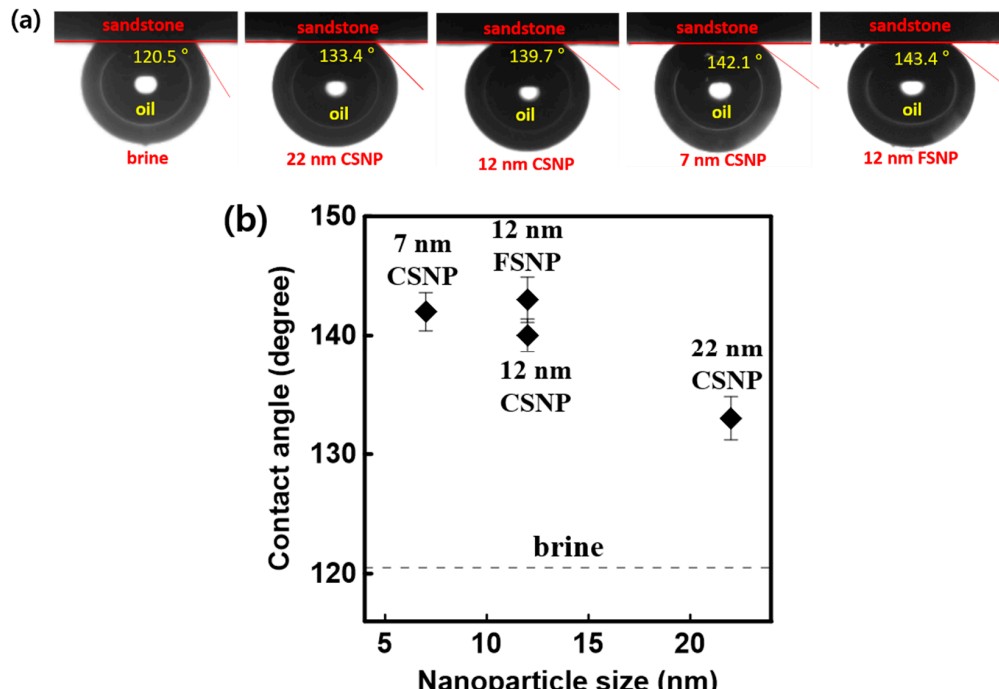

**Figure 7.** (**a**) Contact angles of differently sized NPs and (**b**) contact angles as a function of NP size.

As the equation shows, smaller particles exhibit more Brownian motion, thus enhancing NP absorption onto the rock surface with the maintenance of between-particle spacing. Second, the NP number increases as particle size decreases at a fixed particle concentration; the CSNPs were of similar density ($2.36 \times 10^{-21}$ to $2.50 \times 10^{-21}$ g/nm$^3$). More particles adsorb to the rock surface creating a wedge film between the oil and the rock, thereby changing the contact angle on the rock surface. In theory, the disjoining pressure is the pressure required to remove fluid from the surface of a reservoir rock, thereby overcoming the adhesion force [12,38]. The nanofluid film exerts a structural disjoining force between the oil and the rock surface because of an increase in NP entropy in the nanofluid, reducing oil adsorption to the rock surface. The oscillation amplitude of the film energy increases as the particle diameter decreases because more particles are pumped into the film by the entropic forces created by the confinement effect of the film [12,39,40]. Indeed, we found that the 12 nm FSNP dispersion exhibited the highest contact angle because it has the highest number of NPs (of all NPs tested) in the aqueous phase, although the difference in the contact angle was negligible compared with the difference of the 7 nm CSNP dispersion. These results are similar to the adsorption phenomenon of surfactant on the rock surface. Adsorption density increased with the increase of surfactant concentrations [41]. In the case of NP, the smaller the particle size, the higher the number of particles, thereby the adsorption density on the rock surface may increase with smaller NPs. Consequently, particle size affected the number of particles in the aqueous phase, which affected the contact angle with the rock surface.

### 3.5. Interfacial Rheological Behaviors of Complex Colloidal Dispersions

To explore how particle size affected interfacial rheological behavior, we investigated changes in viscosity as a function of shear rate using silica NPs with diameters of 7, 12, and 22 nm (Figure 8). All NPs exhibited non-Newtonian shear thinning; the viscosity decreased as the shear rate increased. At low shear, the viscosity increased with decreasing CSNP size. This imply that particle number affected viscosity; the number of CSNPs was higher as decreasing particle, increasing the flow resistance imparted by particle–particle interaction. For the 12 nm FSNP dispersion, the NP number was highest in the aqueous phase because of the low density, associated with many between-NP collisions imparting a higher viscosity under applied shear stress. However, the shear stress on a particle increased with increasing shear rate; interactions among NPs reduced rheological behavior. Thus, nanofluid viscosity remained stable. Similarly, in NP-stabilized emulsions, the number of oil droplets in the emulsion phase influenced the viscosity at low shear rates (approximately 5/s), presumably reflecting collisions between emulsion droplets. However, the emulsion viscosities changed similarly as the shear rate increased, such that they became near-identical at or above 100/s.

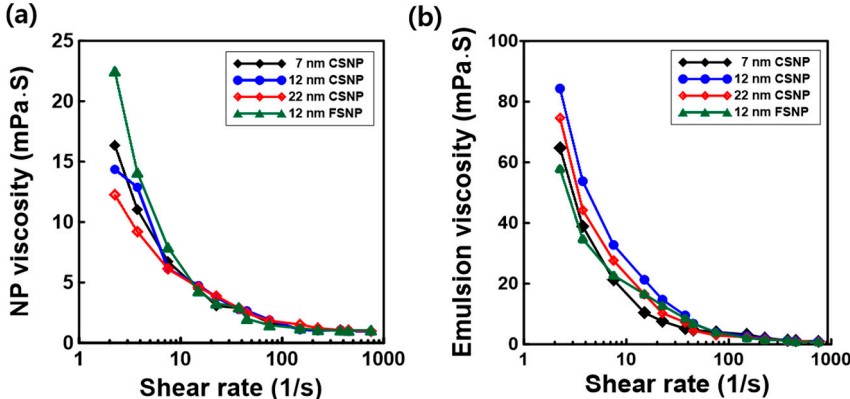

**Figure 8.** Apparent viscosities of (**a**) NPs and (**b**) NP-stabilized emulsions as a function of shear rate.

### 3.6. Effect of NP Number Density on the Oil–Water IFT

Core flooding of Berea sandstone was conducted using brine and silica NP dispersions of different diameters (7 nm CSNP, 12 nm CSNP and FSNP, and 22 nm CSNP). All NPs used were identically surface-modified with PSS-*co*-MA and a zwitterionic surfactant. Oil recovery after the injection of silica NPs was greater than the recovery afforded by brine flooding. Oil recovery increased with decreasing particle size; the recovery factor was 78.2% for the 7 nm CSNP dispersion, 75.1% for the 12 nm CSNP dispersion, and 74.2% for the 22 nm CSNP dispersion (Figure 9). These findings may reflect the increase in total NP surface area with decreasing particle size at a fixed particle concentration, such that the NP area in contact with surfactant increases. NP size affected the IFT reduction and the contact angle (Figures 6 and 7). Injection of a 12 nm FSNP dispersion afforded poorer recovery, compared with the CSNP dispersions, thereby reflecting FSNP aggregation that compromised oil recovery. The results of the two FSNP injection experiments showed a significant difference in the recovery factor, although it was performed under the same conditions; the recovery factor of the first experiment is much lower than the second experiment. This may be because of the difference in oil trapping caused by the occurrence of FSNP aggregation. Once massive pore blockage occurs and permeability declines, solid particles can no longer penetrate the formation due to narrowed pores and a buildup of external porous media [42]. In some cases, excessive NP aggregation may cause maximized permeability reduction and it can cause oil to be trapped inside the pore. Based on all the results of the experiments, NP size and type (CSNP or FSNP) affected the oil/water IFT and rock surface wettability, which influenced ultimate oil recovery.

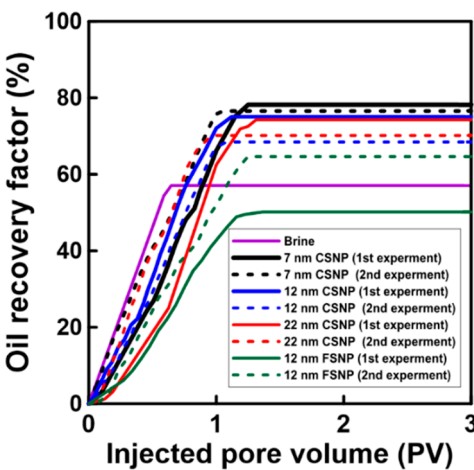

**Figure 9.** Cumulative oil recovery as a function of the injected pore volumes of various fluids.

## 4. Conclusions

We found that the size and type (CSNP or FSNP) of silica NPs affected IFT, rock surface wettability, and rheological behavior. At a fixed particle concentration, the numbers of smaller particles were greater, enhancing the oil–water interfacial properties. Smaller particles more effectively decreased the IFT because the NP–NP spacing was minimal. In addition, smaller particles more strongly adsorbed to the rock, creating a wedge film between the oil and the rock, which changed the oil contact angle. Small particles enhanced viscous behavior by imparting more flow resistance, attributable to particle–particle interaction. Core flooding showed that the oil recovery increased with decreasing particle size. However, FSNP dispersions were less effective than CSNP dispersions as the FSNPs aggregated. Careful selection of NPs markedly enhances oil recovery from rock.

**Author Contributions:** Original draft manuscript, H.A.S.; supervision and review writing, T.L. Both authors have read and agreed to the published version of the manuscript.

**Funding:** This work was supported by the National Research Foundation of Korea (NRF) grant funded by the Korean government (MSIP) (NRF-2019R1F1A1056632). This work was also supported by the research fund of the project of the Korea Institute of Geo-science and Mineral Resources (GP2020-006).

**Institutional Review Board Statement:** Not applicable.

**Informed Consent Statement:** Not applicable.

**Data Availability Statement:** The data presented in this study are available upon request from the corresponding author.

**Conflicts of Interest:** The authors declare no conflict of interest.

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
