# Peer review of "Enhanced Oil Recovery with Size-Dependent Interactions of Nanoparticles Surface-Modified by Zwitterionic Surfactants"

_applsci, doi:10.3390/app11167184_

Round 1
Reviewer 1 Report
The following issues should be modified.
1- Abstract section should be rewritten more scientifically
2- The following references are recommended to cite and discuss
- Parametric study of polymer-nanoparticles-assisted injectivity performance for axisymmetric two-phase flow in EOR processes
- Impact of anionic and cationic surfactants interfacial tension on the oil recovery enhancement
- Effect of anionic and non-anionic surfactants on the adsorption density
3- Literature review is really poor and should be compared to the novelty of your work one by one. For example, say ...et al... did this and we present this. why is different and what are the novelties. It is not acceptable to say just the novel points without any comparison.
4- page 3 line 122, do the materials be stable in higher temperatures? did you mention 25C3? What is it?
5- The quality of the Figures are not good enough. I suggest redrawing them with other software
6- Please add IFT tests and their measurements?
Author Response
We would like to thank the reviewer for a very thorough reading of the manuscript and helpful comments. We have made changes to the manuscript in response, as the attached word file.

Reviewer 2 Report
Line 122: What is 25°C.3? Is that a typo?
Line 129: Please add the diagram of your core flooding system.
Line 141: What is the pressure you applied to the downstream?
Line 159: I understand the sequence for figures is based on the particle size. While I will recommend re-organizing the figures with CSNPS first (a,b,c) and FSNPs (d). Or you can separate those two types of NPs into two parts with individual titles under.
Why you just analyzed the FNSP with 12nm?
Line 285: We name it as the “recovery factor” while the “recovery rate” is rare.
Line 286-287: You may consider adding a table beside figure 8 to present the recovery factors.
Line 295: I noticed that you performed two flooding tests with 12nm FSNPs. The results from these two experiments are not matched well. Please explain the potential reason. Were these two tests performed under different conditions? It is worth having a further and deep discussion on this topic.
Normally, the reservoir conditions include high temperature, high pressure, and salinity. Will you have a further research plan or already have the research results investigated under those conditions? Especially when performing the core flooding experiments.
Author Response

(The authors gave the same response as above.)

Round 2
Reviewer 2 Report
The authors covered my comments and I recommend publishing this manuscript